# Latent Plan Transformer for Trajectory Abstraction: Planning as Latent Space Inference

**Deqian Kong**[1,*]**, Dehong Xu**[1,*]**, Minglu Zhao**[1,*]**, Bo Pang**[2]**, Jianwen Xie**[3]**,**
**Andrew Lizarraga**[1]**, Yuhao Huang**[4]**, Sirui Xie**[5,*]**, Ying Nian Wu**[1]

[1]Department of Statistics and Data Science, UCLA
[2]Salesforce Research [3]Akool Research [4]Xi'an Jiaotong University
[5]Department of Computer Science, UCLA

## Abstract

In tasks aiming for long-term returns, planning becomes essential. We study generative modeling for planning with datasets repurposed from offline reinforcement learning. Specifically, we identify temporal consistency in the absence of step-wise rewards as one key technical challenge. We introduce the Latent Plan Transformer (LPT), a novel model that leverages a latent variable to connect a Transformer-based trajectory generator and the final return. LPT can be learned with maximum likelihood estimation on trajectory-return pairs. In learning, posterior sampling of the latent variable naturally integrates sub-trajectories to form a consistent abstraction despite the finite context. At test time, the latent variable is inferred from an expected return before policy execution, realizing the idea of *planning as inference*. Our experiments demonstrate that LPT can discover improved decisions from sub-optimal trajectories, achieving competitive performance across several benchmarks, including Gym-Mujoco, Franka Kitchen, Maze2D, and Connect Four. It exhibits capabilities in nuanced credit assignments, trajectory stitching, and adaptation to environmental contingencies. These results validate that latent variable inference can be a strong alternative to step-wise reward prompting.

## 1 Introduction

Decision Transformer (DT) (Chen et al., 2021) and some concurrent work (Janner et al., 2021) have popularized the research agenda of decision-making via generative modeling. The general idea is to consider decision-making as a generative process that takes in a representation of the task objective, e.g. the rewards or returns of a trajectory, and outputs a representation of the trajectory. Intuitively, a purposeful decision-making process should shift the trajectory distribution towards regimes with higher returns. In the classical decision-making literature, this is achieved by two interweaving processes, policy evaluation and policy improvement (Sutton and Barto, 2018). Policy evaluation promotes consistency in the estimated correlations between the trajectories and the returns. In DT, this is realized by the maximum likelihood estimation (MLE) of the joint distribution of sequences consisting of states, actions, and return-to-gos (RTG). Policy improvement shifts the distribution to improve the status quo expectation of the returns. In DT, this is naturally entailed since the policy is a distribution of actions conditioned on step-wise RTGs.

In this work, we are interested in the problem of *planning*. Among various ways to identify *planning* as a special class of decision-making problems, we pay particular attention to its data specification

---

*Equal Contribution

Project page: https://sites.google.com/view/latent-plan-transformer.
Code: https://github.com/mingluzhao/Latent-Plan-Transformer.

and inductive biases. As designing step-wise rewards requires significant effort and domain expertise, we focus on the problem of learning from trajectory-return pairs, where a trajectory is a sequence of states and actions, and the return is its total rewards. This design choice forces the agents to predict into the long-term future and figure out step-wise credits by themselves. A competitive Temporal Difference (TD) learning baseline, CQL (Kumar et al., 2020), was reported to be fragile under this data specification (Chen et al., 2021).

Our design of inductive biases reflects our intuition of a *plan*. While a policy is a factor of the trajectory distribution, a *plan* is an abstraction lifted from the space of trajectories. As a plan is always made in advance of receiving returns, it implies *significance*, *persistence*, and *contingency*. An agent should plan for more significant returns. It should be persistent in its plan even if the return is assigned in hindsight. It should also be adaptable to the environment's changes during the execution of the plan. We formulate this hierarchy of decision-making with a top-down latent variable model. The latent variable we introduce is effectively a *plan*, for it decouples the trajectory generation from the expected improvement of returns. The autoregressive policy always consults this temporally extended latent variable to be persistent in the plan. The top-down structure enables the agent to disentangle the variations in its plan from the environment's contingencies.

In this work, we introduce the Latent Plan Transformer (LPT), a novel generative model featuring a latent vector modeled by a neural transformation of Gaussian white noise, a Transformer-based policy conditioned on this latent vector and a return estimation model. LPT is learned by maximum likelihood estimation (MLE). Given an expected return, posterior inference of the latent vector in LPT is an explicit process for iterative refinement of the *plan*. The inferred latent variable replaces RTG in the conditioning of the auto-regressive policy, providing richer information about the anticipated future. We further develop a mode-seeking sampling scheme that strongly enforce the temporal consistency for long-range planning, which is particularly effective in *stitch* trajectory, i.e., to compose parts of sub-optimal trajectories to reach far beyond (Fu et al., 2020). LPT demonstrates competitive performance in Gym-Mujoco locomotion, Franka kitchen, goal-reaching tasks in maze2d and antmaze, and a contingent planning task Connect Four. These empirical results support that latent variable inference can enable and improve planning in the absence of step-wise rewards.

## 2 Background

A sequential decision-making problem can be formulated with a decision process $\langle S, A, H, Tr, r, \rho \rangle$ that contains a set $S$ of states and a set $A$ of actions. Horizon $H$ is the maximum number of steps the agent can execute before the termination of the sequence. We further employ $S^+$ to denote the set of all non-empty state sequences within the horizon and $A^+$ for action sequences likewise. $Tr : S^+ \times A^+ \mapsto \Pi(S)$ is the transition that returns a distribution over the next state. $r : S^+ \times A^+ \mapsto \mathbb{R}$ specifies the real-valued reward at each step. $\rho : \Pi(S)$ is the initial state distribution that is always uncontrollable to the agent. The agent's decisions follow a policy $\pi : S^+ \times A^+ \mapsto \Pi(A)$. In each episode, the agent interacts with the transition model to generate a trajectory $\tau = (s_1, a_1, s_2, a_2..., s_H, a_H)$.

The objective of sequential decision-making is typically formulated as the expected trajectory return $y = \sum_{t=0}^{H} r_t$, $Q = \mathbb{E}_{p(\tau)}[y]$. Conventional RL algorithms solve for a policy $\pi(a_t|s_t, *)$, where the conditioning $*$ denotes the optimal expected return. DT generalizes this policy to $\pi(a_t|s_{\leq t}, a_{<t}, RTG_{\leq t})$, by fitting the joint distribution $p(s_1, a_1, RTG_1, ...s_T, a_T, RTG_T)$ with a Transformer. $RTG_t$ is the return-to-go from step $t$ to the horizon $H$, $RTG_t = \sum_{k=t}^{H} r(s_{\leq k}, a_{\leq k})$. It is a useful indication of future rewards, especially when rewards are dense and informative.

However, $RTG$ becomes less reliable when rewards are sparse or have non-trivial relations with the return. Distributing the return to each step is a credit assignment problem. Consider an example of an ideal credit assignment mechanism: When students receive partial credits for their incomplete answers, it's more fair to give points equal to the full marks minus the expected points for all possible ways to finish the answer, rather than assuming students have no knowledge of the remaining parts. This credit assignment mechanism can be formalized as, $RTG_t^Q = \sum_{k=t}^{K} r(s_{\leq k}, a_{\leq k}) + \mathbb{E}[Q(s_{\leq K}, a_{\leq K})]$. Here $Q$ can be estimated using deep TD learning with multi-step returns. Yamagata et al. (2023) instantiate a Markovian version and demonstrate improvement in trajectory *stiching*.

Whatever credit assignment we use, be it $RTG$ or $RTG^Q$, the purpose is to explicitly model the statistical association between trajectory steps and final returns. This effort is believed to be necessary because of the exponential complexity of the trajectory space. This belief, however, can be re-examined given the success of sequence modeling. We explore an alternative design choice by directly associating the latent vector that generates the trajectory with the return.

## 3 Latent Plan Transformer (LPT)

### 3.1 Model

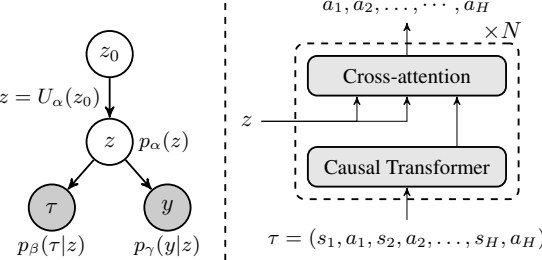

Figure 1: *Left*: Overview of Latent Plan Transformer (LPT). $z \in \mathbb{R}^d$ is the latent vector. The prior distribution of $z$ is a neural transformation of $z_0$, i.e., $z = U_\alpha(z_0)$, $z_0 \sim \mathcal{N}(0, I_d)$. Given $z$, $\tau$ and $y$ are independent. $p_\beta(\tau|z)$ is the trajectory generator. $p_\gamma(y|z)$ is the return predictor. *Right*: Illustration of trajectory generator $p_\beta(\tau|z)$.

Given a variable-length trajectory $\tau$, $z \in \mathbb{R}^d$ is a vector that represents $\tau$ in the latent space. $y \in \mathbb{R}$ is the return of the trajectory. The joint distribution of the trajectory and its return is defined as $p(\tau, y)$.

The latent trajectory variable $z$, conceptualized as a plan, is posited to decouple the autoregressive policy and return estimation. From a statistical standpoint, with $z$ given, we assume that $\tau$ and $y$ are conditionally independent, positioning $z$ as the information bottleneck. Under this assumption, the Latent Plan Transformer (LPT) can be defined as,

$$p_\theta(\tau, y, z) = p_\alpha(z)p_\beta(\tau|z)p_\gamma(y|z), \tag{1}$$

where $\theta = (\alpha, \beta, \gamma)$. LPT approximates the data distribution $p_{\text{data}}(\tau, y)$ using the marginal distribution $p_\theta(\tau, y) = \int p_\theta(\tau, y, z)dz$. It also establishes a generation process,

$$z \sim p_\alpha(z), \quad [\tau|z] \sim p_\beta(\tau|z), \quad [y|z] \sim p_\gamma(y|z). \tag{2}$$

The prior model $p_\alpha(z)$ is an implicit generator, defined as a learnable neural transformation of an isotropic Gaussian, $z = U_\alpha(z_0)$ and $z_0 \sim \mathcal{N}(0, I_d)$. $U_\alpha(\cdot)$ is an expressive neural network, such as the UNet (Ronneberger et al., 2015). This approach is inspired by, yet contrasts with Pang et al. (2020), wherein the latent space prior is modeled as an Energy-based Model (EBM) (Xie et al., 2016). While EBM offers explicit unnormalized density, its sampling process is complex. Conversely, our model provides an implicit density with simpler sampling.

The trajectory generator $p_\beta(\tau|z)$ is a conditional autoregressive model with finite context $K$, $p_\beta(\tau|z) = \prod_{t=1}^{H} p_\beta(\tau_{(t)}|\tau_{(t-K)}, ..., \tau_{(t-1)}, z)$ where $\tau_{(t)} = (s_t, a_t)$. It can be parameterized by a causal Transformer with parameter $\beta$, similar to Decision Transformer (Chen et al., 2021). Specifically, the latent variable $z$ is included in trajectory generation using cross-attention, as shown in Fig. 1 and controls each step of the autoregressive trajectory generation as $p_\beta(a_t|s_{t-K:t}, a_{t-K:t-1}, z)$. The action is assumed to follow a single-mode Gaussian distribution, i.e. $a_t \sim \mathcal{N}(g_\beta(s_{t-K:t}, a_{t-K:t-1}, z), I_{|A|})$.

The return predictor is a non-linear regression on the latent trajectory variable $z$, modeled as $p_\gamma(y|z) = \mathcal{N}(r_\gamma(z), \sigma^2)$. It directly predicts the final return from the latent variable $z$. The function $r_\gamma(z)$ is a small multi-layer perceptron (MLP) that estimates $y$ based on $z$. The variance $\sigma^2$, is treated as the hyper-parameter in our setting.

## 3.2 Offline Learning

With a set of offline training examples $\{(\tau_i, y_i)\}_{i=1}^n$, we aim to learn Latent Plan Transformer (LPT) through maximum likelihood estimation (MLE). The log-likelihood function is defined as $L(\theta) = \sum_{i=1}^n \log p_\theta(\tau_i, y_i)$. The joint probability of the trajectory and final return is

$$p_\theta(\tau, y) = \int p_\beta(\tau|z = U_\alpha(z_0))p_\gamma(y|z = U_\alpha(z_0))p_0(z_0)dz_0, \tag{3}$$

where $p_0(z_0) = \mathcal{N}(0, I_d)$. The learning gradient of log-likelihood can be calculated according to

$$\nabla_\theta \log p_\theta(\tau, y) = \mathbb{E}_{p_\theta(z_0|\tau, y)}[\nabla_\theta \log p_\beta(\tau|U_\alpha(z_0)) + \nabla_\theta \log p_\gamma(y|U_\alpha(z_0))]. \tag{4}$$

The full derivation of the learning method is in Appendix A.1. Let $\delta_\alpha, \delta_\beta, \delta_\gamma$ represent the expected gradients of $L(\theta)$ with respect to the model parameters $\alpha, \beta, \gamma$, respectively. The learning gradients for each component are formulated as follows.

For the prior model $p_\alpha(z)$,

$$\delta_\alpha(\tau, y) = \mathbb{E}_{p_\theta(z_0|\tau, y)}[\nabla_\alpha(\log p_\beta(\tau|z = U_\alpha(z_0)) + \nabla_\alpha \log p_\gamma(y|z = U_\alpha(z_0))].$$

For the trajectory generator,

$$\delta_\beta(\tau, y) = \mathbb{E}_{p_\theta(z_0|\tau, y)}[\nabla_\beta \log p_\beta(\tau|z = U_\alpha(z_0))],$$

For the return predictor,

$$\delta_\gamma(\tau, y) = \mathbb{E}_{p_\theta(z_0|\tau, y)}[\nabla_\gamma \log p_\gamma(y|z = U_\alpha(z_0))].$$

Estimating these expectations requires Markov Chain Monte Carlo (MCMC) sampling of the posterior distribution $p_\theta(z_0|\tau, y)$. We use the Langevin dynamics (Neal, 2011) for MCMC sampling, iterating as follows for a target distribution $\pi(z)$:

$$z^{k+1} = z^k + s\nabla_z \log \pi(z^k) + \sqrt{2s}\epsilon^k, \tag{5}$$

where $k$ indexes the time step of the Langevin dynamics, $s$ is the step size, and $\epsilon^k \sim \mathcal{N}(0, I_d)$ is the Gaussian white noise. Here, $\pi(z)$ is instantiated as the posterior distribution $p_\theta(z_0|\tau, y)$. We have $p_\theta(z_0|\tau, y) \propto p_0(z_0)p_\gamma(y|z)p_\beta(\tau|z)$, where $z = U_\alpha(z_0)$, such that the gradient is

$$\nabla_{z_0} \log p_\theta(z_0|\tau, y) = \nabla_{z_0} \underbrace{\log p_0(z_0)}_{\text{prior}} + \nabla_{z_0} \underbrace{\log p_\gamma(y|z)}_{\text{return prediction}} + \underbrace{\sum_{t=1}^H \nabla_{z_0} \log p_\beta(\tau_{(t)}|\tau_{(t-K:t-1)}, z)}_{\text{aggregating finite-context sub-trajectories}}.$$

This demonstrates that the posterior inference of $z$ is an explicit process of optimizing a plan given its likelihood. In the presence of a finite context, $p_\beta(\tau|z)$ parametrized with Transformer can only account for sub-trajectories with a maximum length of $K$. The latent variable $z$ serves as an abstraction that integrates information from both the final return and sub-trajectories using gradients.

The sampling process starts by initializing $z_0^{k=0}$ from a standard normal distribution $\mathcal{N}(0, I_d)$. We then apply $N$ steps of Langevin dynamics (e.g., $N = 15$) to approximate the posterior distribution, making our learning algorithm an approximate MLE. For a theoretical understanding of this noise-initialized finite-step MCMC, see Pang et al. (2020); Nijkamp et al. (2020); Xie et al. (2023). However, for large horizons (e.g.,$H$=1000), this method becomes slow and memory-intensive. To mitigate this, we adopt the persistent Markov Chain (PMC) (Tieleman, 2008; Xie et al., 2016; Han et al., 2017), which amortizes sampling across training iterations. During training, $z_0^{k=0}$ is initialized from the previous iteration and the number of updates is reduced to $N = 2$ steps. See Appendix A.2 for training and architecture details.

## 3.3 Planning as Inference

The MLE learning of LPT gives us an agent that can plan. During testing, we first infer the latent $z_0$ given the desired return $y$ using Bayes' rule,

$$z_0 \sim p_\theta(z_0|y) \propto p_0(z_0)p_\gamma(y|z = U_\alpha(z_0)). \tag{6}$$

This posterior sampling is achieved using Langevin dynamics similar to the training process. Specifically, we replace the target distribution in Eq. (5) with $p_\theta(z_0|y)$ and run MCMC for a fixed number of steps. Sampling from $p_\theta(z_0|y)$ eliminates the need for expensive back-propagation through the trajectory generator $p_\beta(\tau|z)$.

This posterior sampling of $p(z_0|y)$ is an explicit process that iteratively refines the latent plan $z$, increasing its likelihood given the desired final return. It aligns with our intuition that planning is an inference process. This inferred $z$, fixed ahead of the policy execution, effectively serves as a plan. At each step, the agent consults this plan to generate actions conditioned on the current state and recent history, $a_t \sim p_\beta(a_t|s_{t-K:t-1}, a_{t-K:t-1}, z = U_\alpha(z_0))$.

Once a decision is made, the environment's (possibly non-Markovian) transition $s_{t+1} \sim p(s_{t+1}|a_t, s_t)$ emits the next state. This sequential decision-making process iterates the sampling of $s_t$ and $a_t$ until termination at the horizon.

**Exploitation-inclined Inference (EI)**   Inspired by the classifier guidance (CG) (Dhariwal and Nichol, 2021; Ho and Salimans, 2022) in conditional diffusion models, we introduce a guidance weight $w$ to the original posterior in Eq. (6)

$$\tilde{p}_\theta(z_0|y) \propto p_0(z_0)p_\gamma(y|z)^w, z = U_\alpha(z_0), \tag{7}$$

which has the score $\nabla_{z_0} \log \tilde{p}_\theta(z_0|y) = \nabla_{z_0} \log p_0(z_0) + w\nabla_{z_0} \log p_\gamma(y|z)$. This guidance weight $w$ controls the interpolation between exploration and exploitation. When $w = 1$, the sampled plans collectively represent the posterior density and account for Bayesian uncertainty, resulting in a provably efficient exploration scheme (Osband and Van Roy, 2017). When $w > 1$, the sampled plans are more concentrated around the modes of the posterior distribution, which are plans more likely to the agent. The larger the value of $w$, the more confident the agent becomes, and the stronger the inclination towards exploitation.

An overview of the algorithms for both offline learning and inference can be found in the following.

---

**Algorithm 1** Offline learning

---

**Input:** Learning iterations $T$, initial parameters $\theta_0 = (\alpha_0, \beta_0, \gamma_0)$, offline training samples $\mathcal{D} = \{\tau_i, y_i\}_{i=1}^n$, posterior sampling step size $s$, the number of steps $N$, and the learning rate $\eta_0, \eta_1, \eta_2$.
**Output:** $\theta_T$
**for** $t = 1$ **to** $T$ **do**
    1.**Posterior sampling**: For each $(\tau_i, y_i)$, sample $z_0 \sim p_{\theta_t}(z_0|\tau_i, y_i)$ using Eq. (5) with $N$ steps and step-size $s$, where the target distribution $\pi$ is $p_{\theta_t}(z_0|\tau_i, y_i)$.
    2.**Learn prior model** $p_\alpha(z)$, **trajectory generator** $p_\beta(\tau|z)$ and **return predictor** $p_\gamma(y|z)$:
    $\alpha_{t+1} = \alpha_t + \eta_0 \frac{1}{n}\sum_i \delta_\alpha(\tau_i, y_i); \beta_{t+1} = \beta_t + \eta_1 \frac{1}{n}\sum_i \delta_\beta(\tau_i, y_i); \gamma_{t+1} = \gamma_t + \eta_2 \frac{1}{n}\sum_i \delta_\gamma(\tau_i, y_i)$
    as in Section 3.2.
**end for**

---

**Algorithm 2** Planning as inference

---

**Input:** Expected return $y$, a trained model on offline dataset $\theta$, posterior sampling step size $s$ and the number of steps $N$, Horizon $H$ and an evaluation environment.
**Output:** Trajectory $\tau$
**if** Exploitation-inclined Inference (EI) **then**
    Sample $z_0 \sim \tilde{p}_\theta(z_0|y)$ as in Eq. (7) using Eq. (5) with $N$ steps and step size $s$, where the target distribution $\pi$ is replaced by $\tilde{p}_\theta(z_0|y) \propto p_0(z_0)p_\gamma(y|z = U_\alpha(z_0))^w$ and $z = U_\alpha(z_0)$.
**else**
    Sample $z_0 \sim p_\theta(z_0|y)$ as in Eq. (6) using Eq. (5) with $N$ steps and step size $s$, where $\pi$ is replaced by $p_\theta(z_0|y) \propto p_0(z_0)p_\gamma(y|z = U_\alpha(z_0))$ and $z = U_\alpha(z_0)$.
**end if**
**while** current time step $t \leq H$ **do**
    Sample $a_t$ using trajectory generator as $a_t \sim p_\beta(a_t|s_{t-K:t-1}, a_{t-K:t-1}, z = U_\alpha(z_0))$.
    Once a decision is made, the environment's transition $s_{t+1} \sim p(s_{t+1}|a_t, s_t)$ emits the next state.
**end while**

---

# 4    A Sequential Decision-Making Perspective

We approach the sequential decision-making problem with techniques from generative modeling. In particular, our data specification of trajectory-return pairs omits step-wise rewards, based on the belief that the step-wise reward function is only a proxy of the trajectory return. However, step-wise rewards are indispensable input to classical decision-making algorithms. Accumulating the rewards from the current step to the future gives us the $RTG$, which naturally hints the future progress of the trajectory. How is temporal consistency enforced in our model without the assistance of the $RTG$s?

Without loss of generality, consider the trajectory distribution conditioned on a single return value $y$. The MLE objective is equivalent to minimizing the KL divergence between the data distribution and model distribution, $D_{\mathrm{KL}}(p_{\mathcal{D}}^y(\tau)\|p_\theta^y(\tau))$. Here, $p_{\mathcal{D}}$ denotes the data distribution and $p_\theta$ denotes the model distribution. MLE upon autoregressive modeling imposes additional inductive biases by transforming the objective to $D_{\mathrm{KL}}(p_{\mathcal{D},\mathrm{AR}}^y(\tau)\|p_{\theta,\mathrm{AR}}^y(\tau))$, which is reduced to next-token prediction for behavior cloning and transition model estimation:

$$\underbrace{\sum_{t=1}^{H} D_{\mathrm{KL}}(p_{\mathcal{D}}^y(a_t|s_{1:t},a_{1:t-1})\|p_\theta^y(a_t|s_{1:t},a_{1:t-1}))}_{\text{behavior cloning}}+\underbrace{\sum_{t=1}^{H} D_{\mathrm{KL}}(p_{\mathcal{D}}^y(s_{t+1}|s_{1:t},a_{1:t})\|p_\theta^y(s_{t+1}|s_{1:t},a_{1:t}))}_{\text{transition model estimation}}.$$

However, behavior cloning is believed to suffer from drifting errors since it ignores *covariate shifts* in future steps (Ross and Bagnell, 2010). This concern is unique to sequential decision-making, as the agent cannot control the next state from a stochastic environment, like generating the next text token.

This temporal consistency issue could be alleviated by additionally modeling the sequence of $RTG$. Denote $\rho = (RTG_0, RTG_1, ...RTG_H)$. Modeling the joint distribution is to minimize

$$D_{\mathrm{KL}}(p_{\mathcal{D}}^y(\tau,\rho)\|p_\theta^y(\tau,\rho)) = D_{\mathrm{KL}}(p_{\mathcal{D}}^y(\tau)\|p_\theta^y(\tau)) + D_{\mathrm{KL}}(p_{\mathcal{D}}^y(\rho|\tau)\|p_\theta^y(\rho|\tau))$$

$$=D_{\mathrm{KL}}(p_{\mathcal{D},\mathrm{AR}}^y(\tau)\|p_{\theta,\mathrm{AR}}^y(\tau)) + \mathbb{E}_{p_{\mathcal{D}}^y(\tau)}[\sum_{t=1}^{H} \underbrace{D_{\mathrm{KL}}(p_{\mathcal{D}}^y(RTG_t|\tau)\|p_\theta^y(RTG_t|\tau))}_{\text{RTG prediction}}]. \qquad (8)$$

Note that the *RTG prediction* term is conditioned on the entire trajectory, including the future steps. Minimizing this additional KL divergence correlates predicted $RTG$s with hindsight trajectory-to-go.

Our modeling of the latent trajectory variable $z$ provides an alternative solution to the temporal consistency issue. Eq. (4) is minimizing the KL divergence

$$D_{\mathrm{KL}}(p_{\mathcal{D}}^y(\tau,z)\|p_\theta^y(\tau,z)) = D_{\mathrm{KL}}(p_{\mathcal{D}}^y(\tau)\|p_\theta^y(\tau)) + D_{\mathrm{KL}}(p_{\bar\theta}^y(z|\tau)\|p_\theta^y(z|\tau))$$

$$=D_{\mathrm{KL}}(p_{\mathcal{D},\mathrm{AR}}^y(\tau)\|p_{\theta,\mathrm{AR}}^y(\tau)) + \mathbb{E}_{p_{\mathcal{D}}^y(\tau)}[\underbrace{D_{\mathrm{KL}}(p_{\bar\theta}^y(z|\tau)\|p_\theta^y(z|\tau))}_{\text{plan prediction}}], \qquad (9)$$

where $p_{\bar\theta}^y(z|\tau) = p_{\mathcal{D}}^y(\tau,z)/p_{\mathcal{D}}^y(\tau)$ and $\bar\theta = \theta$ highlights these distributions have the same parameterization as $p_\theta^y$ but are wrapped with `stop_grad()` operator when calculating gradients for $\theta$ (Han et al., 2017). Comparing Eqs. (8) and (9), it is now clear that $z$ plays a similar role as $RTG$ in promoting temporal consistency in autoregressive models. Uniquely, $p_{\bar\theta}^y(z|\tau)$ is the temporal abstraction intrinsic to the model, in contrast to step-wise rewards. From a sequential decision-making perspective, $z$ is effectively a *plan* that the agent is persistent to. From a generative modeling perspective, $z$ from different trajectory modes would decompose the density $p^y(a_t|s_{0:t}, a_{0:t-1})$, relieving the burden of learning the autoregressive policy $p_\beta(a_t|s_{0:t}, a_{0:t-1}, z)$.

One caveat is that the *transition model estimation* should not be conditioned on $y$. Mixing up more trajectory regimes could provide additional regularization for its estimation and generalization. Actually, environment stochasticity is a more concerning issue for autoregressive *behavior cloning*, as highlighted by Yang et al. (2022); Paster et al. (2022); Štrupl et al. (2022); Brandfonbrener et al. (2022); Villaflor et al. (2022); Eysenbach et al. (2022). Among them, Yang et al. (2022) pinpoints the issue by viewing $RTG$s as deterministic latent trajectory variables, closely related to what we present here. Uniquely, the latent variable $z$ in our model is inherently multi-modal (hence very non-deterministic) and ignorant of step-wise rewards. We postulate that the overfitting issue might be mitigated. This is validated by our empirical study inspired by Paster et al. (2022).

Although *RTG prediction* and *plan prediction* both promote temporal consistency, they function very differently when mixing trajectories from multiple return-conditioned regimes. *RTG prediction* is a

supervised learning over the joint distribution $p_{\mathcal{D}}(\tau, \rho)$. Simply mixing trajectories from multiple regimes can't encourage generalization to trajectories that are *stitched* with those in the dataset. Yamagata et al. (2023) propose to resolve this by replacing $RTG$ with $RTG^Q$. Intuitively, this augments the distribution $p_{\mathcal{D}}(\tau, \rho)$ with $p_{\mathcal{D}}(\tau', \rho^Q)$, where $\tau'$ denotes trajectories covered by the offline dynamic programming, such as Q learning, and $\rho^Q = (RTG_0^Q, RTG_1^Q, ...Q_H)$. It significantly improves tasks requiring trajectory *stitching*. Conversely, *plan prediction* is an unsupervised learning as it samples from $p_{\mathcal{D}}(\tau, y)p_{\bar{\theta}}(z|\tau, y)$. As $z$ contains more trajectory-related information than step-wise $RTG$s, trajectories lying outside of $p_{\mathcal{D}}(\tau, \rho)$ may be in-distribution for $p_{\mathcal{D}}(\tau, y)p_{\bar{\theta}}(z|\tau, y)$. The return prediction training further shapes the representation of $z$, which can be benefited from denser coverage of $y$. With more return values covered, we may count on neural networks' strong interpolation capability to shift the trajectory distribution with $y$-conditioning.

## 5 Related work

**Decision-Making via Sequence Modeling** Chen et al. (2021) propose Decision Transformer (DT), pioneering this paradigm shift. Concurrently, Janner et al. (2021) explore beam search upon the learned Transformer for model-based planning and inspired later work that searches over the latent state space (Zhang et al., 2022). Lee et al. (2022) report DT's capability in multi-task setting. Zheng et al. (2022) explore the online extension of DT. Yamagata et al. (2023) augment the Monte Carlo RTG in DT with a Q function and show improvement in tasks requiring trajectory *stitching*. Janner et al. (2022) explore diffusion models (Ho et al., 2020) as an alternative generative model family for decision-making. Our model differentiates from all above in data specification and model formulation.

**Latent Trajectory Variables in Behavior Cloning** Yang et al. (2022); Paster et al. (2022) investigate the DT's overfitting to environment contingencies and propose latent variable solutions. Our model is closely related to theirs but unique in an EM-style algorithm for MLE. Ajay et al. (2021); Lynch et al. (2020) propose latent variable models to make Markovian policies temporally extended. Their models are more related to VAE (Kingma and Welling, 2014).

**Offline Reinforcement Learning** Since the offline static datasets only partially cover the state transition spaces, efforts from a conventional RL perspective focus on imposing pessimistic biases to value iteration (Kumar et al., 2020; Kostrikov et al., 2021; Uehara and Sun, 2021; Xie et al., 2021; Cheng et al., 2022). Fujimoto and Gu (2021) show that simply augmenting value-based methods with behavior cloning achieves impressive performance. Emmons et al. (2021) report that supervised learning on return-conditioned policies is competitive to value-based methods in offline RL. Our MLE objective is more related to the supervised learning methods. The latent variable inference further imposes temporal consistency, acting as a replacement of value iteration.

**Hierarchical RL** Methods like OPAL (Ajay et al., 2021), OPOSM (Freed et al., 2023) address TD-learning's limitations in long-range credit assignment using a two-stage approach: discovering skills from shorter subsequences to reduce the planning horizon, then applying skill-level CQL or online model-based planning on the reduced horizons. This paper focuses on comparing various methods for long-range credit assignment on the original horizon. Future work includes first discovering skills and then modeling them with a skill-level LPT to further extend the effective horizon.

## 6 Experiments

The data specification of trajectory-return pairs distinguishes our empirical study from most existing works in offline RL. Omitting step-wise rewards naturally increases the challenges in decision-making.

### 6.1 Overview

Our empirical study adopts the convention from offline RL. We first train our model with the offline data and then test it as an agent in the corresponding task. More training details and ablation studies of LPT can be found in Appendices A.2 and A.4.

**OpenAI Gym-Mujoco** The D4RL offline RL dataset (Fu et al., 2020) features densely-rewarded locomotion tasks including *Halfcheetah, Hopper*, and *Walker2D*. We test for *medium* and *medium-replay*. It also includes *Antmaze*, a locomotion and goal-reaching task with extremely sparse reward.

The agent will only receive a reward of 1 if hitting the target location and 0 otherwise. We use its *umaze* and *umaze-diverse* variants.

**Franka Kitchen** Franka Kitchen is a multitask environment where a Franka robot with nine degrees of freedom operates within a kitchen setting, interacting with household objects to achieve specific configurations. Our experiments focus on two datasets of the environment: *mixed*, and *partial*, which consists of non-task-directed demonstrations and partially task-directed demonstrations respectively.

**Maze2D** Maze2D is a navigation task in which the agent reaches a fixed goal location from random starting positions. The agent is rewarded 1 point when it is around the goal. Experiments are conducted on three layouts: *umaze, medium*, and *large*, with increasing complexity. The training data of the Maze2D task contains only suboptimal trajectories from and to randomly selected locations.

**Connect Four** This is a tile-based game, where the agent plays against a stochastic opponent (Paster et al., 2022), receiving at the end of an episode 1 reward for winning, 0 for a draw, and -1 for losing.

**Baselines** We compare the performance of LPT with several representative baselines including CQL (Kumar et al., 2020), DT (Chen et al., 2021) and QDT (Yamagata et al., 2023). CQL baseline results are obtained from Kumar et al. (2020). QDT baseline results are from Yamagata et al. (2023). The DT results for Gym-Mujoco and Maze2D tasks are from Yamagata et al. (2023), Antmaze from Zheng et al. (2022), and Kitchen implemented based on the published source code. CQL and DT results in the Connect Four experiments are from Paster et al. (2022). The mean and standard deviation of our model, shown as LPT and LPT-EI, are reported over 5 seeds.

Table 1: Evaluation results of offline OpenAI Gym MuJoCo tasks. We provide results for data specification with step-wise reward (left) and final return (right). **Bold** highlighting indicates top scores. LPT outperforms all final-return baselines and most step-wise-reward baselines.

| Dataset | Step-wise Reward | | | Final Return | | | | |
| --- | --- | --- | --- | --- | --- | --- | --- | --- |
| | CQL | DT | QDT | CQL | DT | QDT | LPT (Ours) | LPT-EI (Ours) |
| halfcheetah-medium | **44.4** | 42.1 | 42.3 | 1.0 | 42.4 | 42.4 | $43.13 \pm 0.38$ | **43.53** $\pm 0.08$ |
| halfcheetah-medium-replay | **46.2** | 34.1 | 35.6 | 7.8 | 33.0 | 32.8 | $39.64 \pm 0.83$ | **40.66** $\pm 0.12$ |
| hopper-medium | 58.0 | 60.3 | 66.5 | 23.3 | 57.3 | 50.7 | $58.52 \pm 1.92$ | **63.83** $\pm 1.47$ |
| hopper-medium-replay | 48.6 | 63.7 | 52.1 | 7.7 | 50.8 | 38.7 | $82.29 \pm 1.26$ | **89.93** $\pm 0.61$ |
| walker2d-medium | 79.2 | 73.3 | 67.1 | 0.0 | 69.9 | 63.7 | $77.85 \pm 3.18$ | **81.15** $\pm 0.33$ |
| walker2d-medium-replay | 26.7 | 60.2 | 58.2 | 3.2 | 51.6 | 29.6 | $72.31 \pm 1.92$ | **75.68** $\pm 0.34$ |
| kitchen-mixed | 51.0 | 22.3 | - | - | 17.2 | - | $61.9 \pm 1.22$ | **64.7** $\pm 0.51$ |
| kitchen-partial | 49.8 | 20.4 | - | - | 10.5 | - | $61.2 \pm 1.75$ | **65.3** $\pm 0.62$ |

## 6.2 Credit assignment

When resolving the temporal consistency issue, our model doesn't have an explicit credit assignment mechanism that accounts for the actual contribution of each step. It is not aware of the step-wise rewards either. We are therefore curious about whether the inferred latent variable $z$ can effectively assign fair credits to resolve compounding errors.

**Distributing sparse rewards to high-dimensional actions** The Gym-Mujoco environment was a standard testbed for high-dimensional continuous control during the development of modern RL algorithms (Lillicrap et al., 2015). In this environment, step-wise rewards were believed to be critical for TD learning methods. In the setup of offline RL, Chen et al. (2021) reported the failure of the competitive CQL baseline when delaying step-wise rewards until the end of the trajectories. DT and QDT are reported to be robust to this alternation. As shown in Table 1, the proposed model, LPT, outperforms these baselines when the data specifications are the same. Notably, LPT even excels in most of the control tasks when compared with the baselines with step-wise rewards.

**Distributing delayed rewards to long-range sequences** Maze navigation tasks with fully delayed rewards align with our intuition of a planning problem, for it involves decision-making at certain critical states absent of instantaneous feedback. An ideal planner would take in the expected total return and calculate the sequential decisions, automatically distributing credits from the extremely sparse and fully delayed rewards. According to Yamagata et al. (2023), DT fails in these tasks. Our

Table 2: Evaluation results of Maze2D tasks. **Bold** highlighting indicates top scores.

| Dataset | CQL | DT | QDT | LPT | LPT-EI |
|---|---|---|---|---|---|
| Maze2D-umaze | 5.7 | $31.0 \pm 21.3$ | $57.3 \pm 8.2$ | $65.43 \pm 2.91$ | $\mathbf{70.57 \pm 1.39}$ |
| Maze2D-medium | 5.0 | $8.2 \pm 4.4$ | $13.3 \pm 5.6$ | $20.62 \pm 1.81$ | $\mathbf{26.66 \pm 0.74}$ |
| Maze2D-large | 12.5 | $2.3 \pm 0.9$ | $31.0 \pm 19.8$ | $37.21 \pm 2.05$ | $\mathbf{45.89 \pm 2.98}$ |

proposed model LPT outperforms QDT by a large margin in all three variants of the maze task. These results validate our hypothesis that the additional plan prediction KL imposes temporal consistency on autoregressive policies.

Table 3: Evaluation results of Antmaze tasks. **Bold** highlighting indicates top scores.

| Dataset | CQL | DT | LPT | LPT-EI |
|---|---|---|---|---|
| Antmaze-umaze | 74.0 | $53.3 \pm 5.52$ | $80.8 \pm 4.83$ | $\mathbf{92.4 \pm 0.80}$ |
| Antmaze-umaze-diverse | 84.0 | $52.5 \pm 5.89$ | $78.5 \pm 1.66$ | $\mathbf{84.4 \pm 1.96}$ |

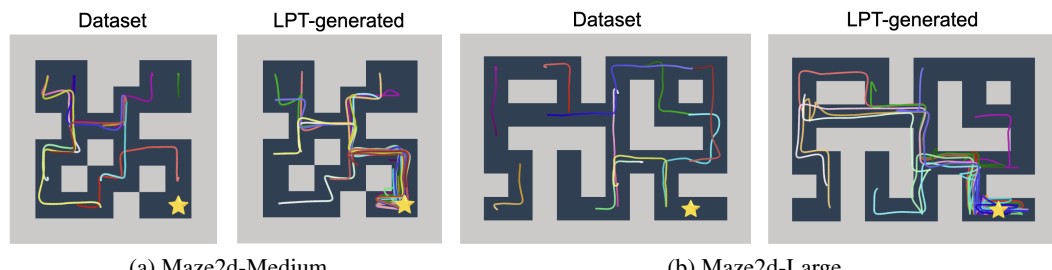

(a) Maze2d-Medium          (b) Maze2d-Large

Figure 2: (a) Maze2D-medium environment (b) Maze2D-large environment. Left panels show example trajectories from the training set and right panels show LPT generations. Yellow stars represent the goal states.

## 6.3 Trajectory *stitching*

In addition to credit assignment, the setup of offline RL further presents a challenge, trajectory *stitching* (Fu et al., 2020), which articulates the problem of shifting the trajectory distribution towards sparsely covered regimes with higher returns. In the Franka Kitchen environment, both the *mixed*, and *partial* datasets contain undirected data where the robot executes subtasks that do not necessarily achieve the goal configuration. The "mixed" dataset contains no complete solution trajectories, necessitating that the agent learn to piece together relevant sub-trajectories. A similar setting happens in Maze2D domain. Taking Maze2D-medium as an example, in the training set, the average return of all trajectories is 3.98 with a standard deviation of 10.44, where the max return is 47. DT's score is only marginally above the average return. Yamagata et al. (2023) attribute DT's failure in Maze2D to its difficulty with trajectory *stitching*.

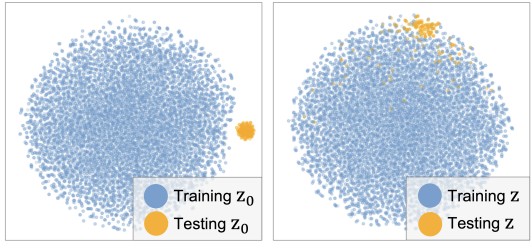

Figure 3: t-SNE plot of latent variables in the Maze2D-medium. Left: Training $z_0$ from aggregated posterior $\mathbb{E}_{p_{\mathcal{D}}(\tau, y)}[p_\theta(z_0|\tau, y)]$. Testing $z_0$ from $p_\theta(z_0|y)$, disjoint from training population. Right: Distribution of $z = U_\alpha(z_0)$.

Fig. 2 visualizes samples from the training data and successful trajectories in testing. The left panels show that trajectories in training are sub-optimal in terms of (1) being short in length and (2) containing very few goal-reaching instances. Trajectories on the right are generated by 10 random runs with LPT, where the agent successfully navigates to the end goal from random starting positions in an effective manner. This indicates that the agent can discover the correlation between different $y$s to facilitate such stitching.

To probe into the agent's understanding of trajectories' returns, we visualize the representation

space of the latent variables. The left of Fig. 3 is the aggregated posterior distribution of $z_0$. We can see that $z_0$ infered from $p_\theta(z_0|y)$ are distant away from the training population. The agent understands they are not very likely in the training set. The right of Fig. 3 is the distribution of $z$, which is transformed from $z_0$ with the UNet, $z = U_\alpha(z_0)$. We observe that $z$s from the generated trajectories become "in-distribution" in the sense that some of them are mingled into the training population and the remaining lie inside a region coverable through linear interpolation of training samples. The agent understands what trajectories to generate even if they are unlikely among what it has seen.

## 6.4 Environment contingencies

To live in a stochastic world, contingent planning that is adaptable to unforeseen noises is desirable. Paster et al. (2022); Yang et al. (2022) discover that DT's performance would degrade in stochastic environments due to inevitable overfitting towards contingencies. We examine LPT and other baselines in Connect Four from Paster et al. (2022). Connect Four is a two-player game, where the opponent will make adversarial moves to deliberately disturb an agent's plan. According to the empirical study from Paster et al. (2022), the degradation of DT is more significant than in stochastic Gym tasks from Yang et al. (2022). As shown in Table 4, LPT achieves the highest score with minimal variance. The ESPER baseline is from Paster et al. (2022), which is very relevant to LPT as it is also a latent variable model. ESPER learns the latent variable model with an adversarial loss. It further adds a clustering loss in the latent space. LPT's on-par performance may justify that MLE upon a more flexible prior can play an equal role.

Table 4: Evaluation results on Connect Four. **Bold** highlighting indicates top scores.

| Dataset | CQL | DT | ESPER | LPT |
|---|---|---|---|---|
| Connect Four | $0.61 \pm 0.05$ | $0.8 \pm 0.07$ | $\mathbf{0.99 \pm 0.03}$ | $\mathbf{0.99 \pm 0.01}$ |

# 7 Limitation

We omit the Antmaze-large experiment from the main text and included potential reasons for LPT's unsatisfactory performance in Appendix A.3. Another interesting direction is to study LPT's continual learning potential. During planning, LPT explores with provably efficient posterior sampling (Osband et al., 2013; Osband and Van Roy, 2017).

# 8 Summary

We study generative modeling for planning in the absence of step-wise rewards. We propose LPT which generates trajectory and return from a latent variable. In learning, posterior sampling of the latent variable naturally gathers sub-trajectories to form an episode-wise abstraction despite finite context in training. In inference, the posterior sampling given the target final return explores the optimal regime of the latent space. It produces a latent variable that guides the autoregressive policy to execute consistently. Across diverse evaluations, LPT demonstrates competitive capacities of nuanced credit assignments, trajectory stitching, and adaptation to environmental contingencies. Contemporary work extends LPT's application to online molecule design (Kong et al., 2024). Future research directions include studying online and multi-agent variants of this model, exploring its application in real-world robotics, and investigating its potential in embodied agents.

## Acknowledgements

The work was partially supported by NSF DMS-2015577, NSF DMS-2415226, and a gift fund from Amazon. We sincerely thank Mr. Shanwei Mu and Dr. Jiajun Lu at Akool Research for their computational support, as well as the anonymous reviewers for their valuable feedback.

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

## A Appendix

### A.1 Details about model and learning

Given a trajectory $\tau$, $z \in \mathbb{R}^d$ is the latent vector to represent the variable-length trajectory. $y \in \mathbb{R}$ is the return of the trajectory. With offline training trajectory-return pairs $\{(\tau_i, y_i), i = 1, ..., n\}$. The log-likelihood function is $L(\theta) = \sum_{i=1}^{n} \log p_\theta(\tau_i, y_i)$, with learning gradient $\nabla_\theta L(\theta) = \sum_{i=1}^{n} \nabla_\theta \log p_\theta(\tau_i, y_i)$. We derive the form of $\nabla_\theta \log p_\theta(\tau_i, y_i)$, proving Eq. (4) below, dropping index subscript $_i$ for simplicity.

$$
\begin{aligned}
\nabla_\theta \log p_\theta(\tau, y) &= \frac{\nabla_\theta p_\theta(\tau, y)}{p_\theta(\tau, y)} \\
&= \frac{1}{p_\theta(\tau, y)} \int \nabla_\theta p_\theta(\tau, y, z = U_\alpha(z_0)) dz_0 \\
&= \int \frac{p_\theta(\tau, y, z = U_\alpha(z_0))}{p_\theta(\tau, y)} \nabla_\theta \log p_\theta(\tau, y, z = U_\alpha(z_0)) dz_0 \\
&= \int p_\theta(z_0 | \tau, y) \nabla_\theta \log p_\theta(\tau, y, z = U_\alpha(z_0)) dz_0 \\
&= \mathbb{E}_{p_\theta(z_0 | \tau, y)} \left[ \nabla_\theta \log p_\theta(\tau, y, z = U_\alpha(z_0)) \right] \\
&= \mathbb{E}_{p_\theta(z_0 | \tau, y)} \left[ \nabla_\theta \log p_\beta(\tau | U_\alpha(z_0)) + \nabla_\theta \log p_\gamma(y | U_\alpha(z_0)) + \nabla_\theta \log p_0(z_0) \right] \\
&= \mathbb{E}_{p_\theta(z_0 | \tau, y)} \left[ \nabla_\theta \log p_\beta(\tau | U_\alpha(z_0)) + \nabla_\theta \log p_\gamma(y | U_\alpha(z_0)) \right].
\end{aligned}
$$

### A.2 Training details

For Gym-Mujoco offline training, as shown in Table 5, most of the hyperparameters were shared across all tasks except context length and hidden size. However, due to the significant variations in the scale of the maze maps and the lengths of the trajectories within the Maze2D environments—spanning umaze, medium, and large categories—model sizes were adjusted accordingly to accommodate these

differences, where the detailed setting can be found in Table 6. We also show the parameters for Franka Kitchen environment in Table 7 and Connect Four in Table 8.

Training time for the Gym-Mujoco tasks using a single Nvidia A6000 GPU is 18 hours on average. We train Maze2d tasks using a single Nvidia A100 GPU using 30 hours on average. Kitchen tasks using a single Nvidia A6000 GPU takes 60 hours on average. Connect-4 on a single Nvidia A6000 GPU takes 10 hours.

Table 5: Gym-Mujoco Environments LPT Model Parameters

| Parameter | HalfCheetah | Walker2D | Hopper | AntMaze |
|---|---|---|---|---|
| Number of layers | 3 | 3 | 3 | 3 |
| Number of attention heads | 1 | 1 | 1 | 1 |
| Embedding dimension | 128 | 128 | 128 | 192 |
| Context length | 32 | 64 | 64 | 64 |
| Learning rate | 1e-4 | 1e-4 | 1e-4 | 1e-3 |
| Langevin step size | 0.3 | 0.3 | 0.3 | 0.3 |
| Nonlinearity function | ReLU | ReLU | ReLU | ReLU |

Table 6: Maze2D Environments LPT Model Parameters

| Parameter | Umaze | Medium | Large |
|---|---|---|---|
| Number of layers | 1 | 3 | 4 |
| Number of attention heads | 8 | 1 | 4 |
| Embedding dimension | 128 | 192 | 192 |
| Context length | 32 | 64 | 64 |
| Learning rate | 1e-3 | 1e-3 | 2e-4 |
| Langevin step size | 0.3 | 0.3 | 0.3 |
| Nonlinearity function | ReLU | ReLU | ReLU |

Table 7: Franka Kitchen Environments LPT Model Parameters

| Parameter | Mixed | Partial |
|---|---|---|
| Number of layers | 4 | 3 |
| Number of attention heads | 4 | 16 |
| Embedding dimension | 128 | 128 |
| Context length | 16 | 16 |
| Learning rate | 1e-3 | 1e-3 |
| Langevin step size | 0.3 | 0.3 |
| Nonlinearity function | ReLU | ReLU |

Table 8: Connect 4 LPT Model Parameters

| Parameter | Value |
|---|---|
| Number of layers | 3 |
| Number of attention heads | 4 |
| Embedding dimension | 128 |
| Context length | 4 |
| Learning rate | 1e-3 |
| Langevin step size | 0.3 |
| Nonlinearity function | ReLU |

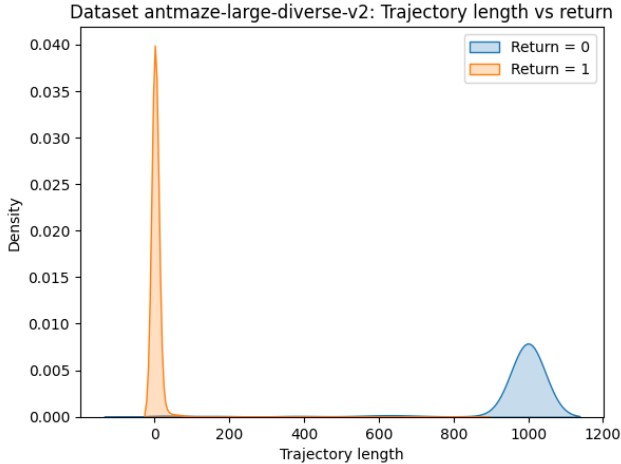

Figure 4: Trajectory length and return distribution in dataset Antmaze-large-diverse

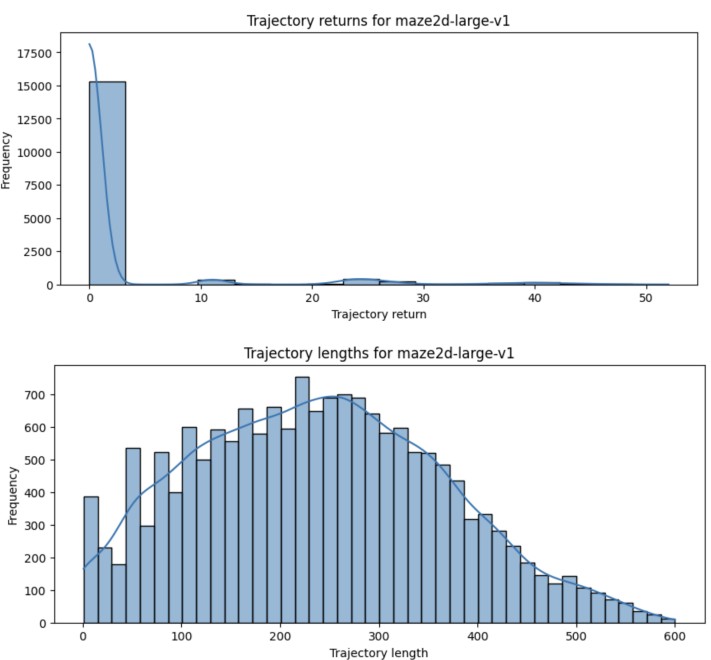

Figure 5: Trajectory length and return distribution in dataset Maze2D-Large

### A.3 Discussion on data quality of Antmaze medium and large

In our experiments, we encounter a curious phenomenon that LPT outperforms CQL, DT and QDT in Antmaze-umaze by a large margin but falls behind in Antmaze-large. Upon closer examination of the data from D4RL, we gained valuable insights into the potential reasons behind LPT's performance on this task.

Fig. 4 plots the distributions of final returns and the trajectory lengths. Surprisingly, this dataset consists of 5448 trajectories (75.86%) with length=1, 893 trajectories (12.43%) with length=1000, and only 841 trajectories (11.71%) with lengths in between. Such a biased trajectory coverage can be detrimental to sequence models like LPT, which learn to make decisions by discovering correlations between trajectories and returns.

As a reference, Fig. 5 shows the distributions of final returns and the trajectory lengths of Maze2D-large, a task where LPT performs well. It is important to note that TD-learning methods, such as CQL and QDT, rely solely on $(s, a, s', r)$ tuples and are less affected by the trajectory length distribution in the dataset. Consequently, Antmaze-large in D4RL remains a fair dataset for these methods to perform offline RL.

## A.4   Ablation study

We investigate the role of the expressive prior $p_\alpha(z)$ in our Latent Plan Transformer (LPT) model by removing the UNet component, which transforms $z_0$ from a non-informative Gaussian distribution. Table 9 reports the results on three Gym-Mujoco tasks and Connect Four. We observe that the performance of LPT drops in all environments when the UNet is removed. For example, in the stochastic environment Connect Four, LPT's performance decreases from 0.99 to 0.90, while the baseline Decision Transformer (DT) without latent variables achieves 0.80. These results indicate that a more flexible prior benefits the learning and inference of LPT.

Table 9: Ablation study results on Gym-Mujoco tasks and Connect Four.

| Dataset | DT | LPT | LPT w/o UNet |
|---|---|---|---|
| halfcheetah-medium-replay | $33.0 \pm 4.8$ | $39.64 \pm 0.83$ | $34.70 \pm 1.58$ |
| hopper-medium-replay | $50.8 \pm 14.3$ | $71.17 \pm 3.01$ | $53.41 \pm 6.95$ |
| walker2d-medium-replay | $51.6 \pm 24.7$ | $72.31 \pm 1.92$ | $56.88 \pm 4.20$ |
| Connect Four | $0.8 \pm 0.07$ | $0.99 \pm 0.01$ | $0.90 \pm 0.06$ |

To further explore the impact of the prior, we conducted additional experiments testing the effects of different UNet configurations on LPT's performance. Table 10 shows the normalized scores on the `walker2d-medium-replay` task with various UNet architectures. We observe that reducing the capacity or expressiveness of the UNet (e.g., smaller dimension, fewer multipliers, smaller initial convolution, or fewer ResBlocks) consistently degrades performance, though still outperforming the model without the UNet prior. This suggests that a more expressive prior enhances LPT's ability to model complex policies.

Table 10: Effect of different UNet configurations on LPT performance.

| Model Prior | Normalized Score |
|---|---|
| UNet (original) | $\mathbf{72.31} \pm 1.92$ |
| UNet w/ smaller dimension | $64.06 \pm 1.94$ |
| UNet w/ fewer multipliers | $64.59 \pm 1.54$ |
| UNet w/ smaller initial convolution | $70.49 \pm 2.84$ |
| UNet w/ single ResBlock | $67.95 \pm 4.64$ |
| No UNet (Standard Normal prior) | $56.88 \pm 4.20$ |

Our results underscore the crucial role of the learned prior in LPT's performance. The original UNet configuration achieves the highest normalized score, indicating that our current UNet design is optimal among the variants tested. We appreciate the reviewer's suggestion, as it prompted us to perform a more detailed analysis of the prior's impact on LPT.

## A.5   Continual learning with online data

We are also interested in LPT's potential in finetuning or even continual learning. Inspired by ODT (Zheng et al., 2022), we employ a trajectory replay butter (Mnih et al., 2015) to store samples from online interaction in a first-in-first-out manner. After the completion of each episode, we update LPT with the same learning algorithm as with the offline data. Note that ODT introduces some techniques additional to DT. In contrast, LPT explores with the provably efficient posterior sampling (Osband et al., 2013; Osband and Van Roy, 2017). We report the results in Table 11. Despite the significance in a few tasks, the improvement is within 1 standard deviation of the mean for the majority. We observe a similar pattern in ODT (Zheng et al., 2022).

Table 11: Evaluation results of online Open AI Gym MuJoCo and Antmaze tasks. ODT baselines are sourced from Zheng et al. (2022). Our results are reported over 5 seeds.

| Dataset | Step-wise Reward | | Final Return | |
| --- | --- | --- | --- | --- |
| | ODT | $\Delta$ | LPT | $\Delta$ |
| halfcheetah-medium | $42.16 \pm 1.48$ | $-0.56$ | $43.26 \pm 0.59$ | $0.13$ |
| halfcheetah-medium-replay | $40.2 \pm 1.61$ | $0.43$ | $40.63 \pm 0.28$ | $0.99$ |
| hopper-medium | $97.54 \pm 2.1$ | $30.59$ | $64.84 \pm 10.29$ | $6.32$ |
| hopper-medium-replay | $88.89 \pm 6.33$ | $2.25$ | $72.44 \pm 8.07$ | $1.27$ |
| walker2d-medium | $76.79 \pm 2.30$ | $4.60$ | $79.54 \pm 5.11$ | $1.69$ |
| walker2d-medium-replay | $76.86 \pm 4.04$ | $7.94$ | $78.99 \pm 3.84$ | $6.68$ |
| Antmaze-umaze | $88.5 \pm 5.88$ | $35.4$ | $83.5 \pm 3.28$ | $22.3$ |
| Antmaze-diverse | $56.00 \pm 5.69$ | $5.8$ | $75.6 \pm 2.8$ | $8.0$ |

## A.6 Broader Impact

We introduce a novel generative model that leverages the idea of "planning as inference", which potentially leads to more capable and efficient AI systems that make better decisions. One potential negative impact could come from the misuse of the generative modeling in designing objects or entities for harmful purposes. Developing suitable safeguards and regulations will be crucial to mitigate potential negative impacts.

