# OpenReview forum: "Latent Plan Transformer for Trajectory Abstraction: Planning as Latent Space Inference"
_NeurIPS.cc/2024/Conference — NeurIPS 2024 poster_

### Official Review · Reviewer_W2Dk · 2024-06-23

**Soundness:** 2
**Presentation:** 2
**Contribution:** 2
**Rating:** 6
**Confidence:** 3

**Summary:**

Authors propose a novel method Latent Plan Transformer (LPT). The key idea is to modify Decision Transformer (DT) approach by adding latent variable conditioning instead of return-to-go. This latent variable is assumed to represent a "plan" that the agent will follow. The motivation of replacing return-to-go is the fact that in practice return-to-go might be unavailable during the inference.

**Strengths:**

To my prior knowledge, authors provide a novel idea of usage of latent variable for decision making with autoregressive generative model.

Authors provide strong empirical results on a diverse range of tasks which show that the proposed method has much higher performance than other types of the DT.

**Weaknesses:**

There is a chance I did not understand some parts of the work which lead to the lack of intuition behind the behavior that we observe.

The problem is that we sample some random variable $z$ that is conditioned on the final return and just plug this fixed into the Transformer's cross-attention (if I understood everything correctly). This provides better results into a better performance than DT even when it has access to the step-wise return to go. It seems very strange as intuitively updated return-to-go should let the model to adapt to the situation in which the agent ended up and change the behavior according to it while fixed latent variable might only provide the direction which the agent should follow and not updated. Can it be the implementation/hyperparameters differences? Baselines scores are taken from previous works and not claimed to be reproduced with LPT codebase and it is not provided. Or is it just some good latent representation of the return-to-go?

Some experimental results are omitted.

**Questions:**

1) What are results of DT/QDT if the same code is used for their training?

2) What is the performance of the LPT using Antmaze medium/large tasks? Umaze tasks are not very representative. It is claimed that performance there is poor and appendix shows that there are some problems with those datasets. Anyway, what are the results? What if we  remove the trajectories of length 1 and train all baselines and LPT using modified datasets?

3) How is the Figure 3 (left) obtained? It is said that $z_0$ is sampled from isotropic normal distribution but why is it in different t-sne space?

4) Ablation: what if we remove the return-to-go conditioning during the training of the U-Net? What if we remove the trajectory conditioning during the training of the U-Net? I find these missing.

5) What kind of distribution over $z$ do we obtain? I would recommend to try some toy example where we have $z$ with dim of 2. Is it  much different from gaussian?

**Limitations:**

The main limitation is the lack of intuition of what is actually happening and lack of latent variable analysis.

---

> ### Author Rebuttal · Authors · 2024-08-07
>
> Thank you for your valuable feedback! We greatly appreciate the opportunity to clarify and expand our work.
> > W1: Lack of intuition: plugging in a sampled fixed random variable conditioned on the final return performs better than DT with step-wise RTG
>
> Thank you for this insightful observation. Indeed, we can view both RTG and the latent variable $z$ as functions of the whole trajectory and final return. RTG is hard-designed, while our latent $z$ with a learnable prior distribution is learned from the posterior distribution $p(z|\tau, y)$. The learned $z$ potentially captures more information than RTGs, serving as a rich, abstract representation of the entire trajectory and expected return, providing guidance throughout the process. In Sec.4, we show that given final return, both latent $z$ and RTG can help promote temporal consistencies. During inference, although RTG updates at each time-step, the non-Markovian state representation at each time-step also consults $z$ to make decisions via the cross-attention mechanism, allowing the model to maintain and utilize information about the trajectory's progress and the remaining challenges.
> > Q1: Results of DT/QDT if the same code is used.
>
> Regarding the implementation of DT, we used the publicly available code provided by Huggingface. For QDT, since the authors did not release a public version of the code, we relied on the results reported in their original paper. We will release our codebase to facilitate reproducibility and further investigation.
> > Q2.1: LPT performance in Antmaze medium/large tasks.
>
> Thanks for pointing out this important aspect of our work, which indeed presents some challenges as we noted in our paper's limitations section. For Antmaze, we attach here the results as the reviewer suggested:
> ||CQL|DT|LPT-EI|
> |-|-|-|-|
> |Antmaze-diverse-large|14.9 ± 3.2|0.0 ± 0.0|3.6 ± 0.8|
>
> These results show that LPT's performance is lower than CQL, though it outperforms DT. The difference in performance highlights an important distinction between trajectory-based models and TD-learning methods in offline reinforcement learning, as we mentioned in the **global response**. Sequence models like LPT and DT are designed to learn from correlations between sequences of actions and states and their associated returns. However, this strength can become a limitation when faced with datasets that have skewed distributions of trajectory lengths, as in the Antmaze-large environment. If the dataset is dominated by very short or very long trajectories, it can bias the model's understanding of this relationship.
> > Q2.2: Removing the trajectories of length 1 and train all baselines and LPT using modified datasets.
>
> This is a very interesting and insightful point! We conduct another set of experiments where we remove trajectories of length 1 from the Antmaze-large-diverse dataset to perform model training. The results are as follows:
> ||DT|LPT-EI|
> |-|-|-|
> |Antmaze-diverse-large|0.0 ± 0.0|3.2±0.9|
>
> Results indicate that the performance of LPT remains suboptimal -- unfortunately upon closer examination of the new dataset, we found that the distribution the trajectory lengths are still biased, as we show in **Fig. 2 in our global response PDF**. We can see that most of the trajectories with final return=1 are very short in length (median = 13). The trajectory coverage can be detrimental to sequence models like LPT, which learn to make decisions by discovering correlations between trajectories and returns. We appreciate the reviewers' constructive feedback, which has prompted us to conduct a thorough analysis. This analysis highlighted the importance of dataset quality for sequence models.
> > Q3: Figure 3: distribution of $z_0$ and t-sne plot
>
> Good catch! In Figure 3 (left), we illustrate the learned distribution in the $z_0$ space. For the blue points, we plot $\mathbb{E}_{p(\tau, y)}p(z_0|\tau,y)$ where each point represents one training data point in $z_0$ space. For yellow points, we plot, in the testing or inference case, the sampled $p(z_0|y^*)$, given expected final return $y^*$. Since $y^*$ is chosen outside the training distribution, these $z_0$s may be distant from training dataset.
> > Q4: Ablation: removing the RTG conditioning during the training of the UNet
>
> Thanks for raising this point! We would like to clarify that in LPT, the UNet is not conditioned on RTG or trajectories during training. Instead, it serves as a flexible prior for the latent variable $z$. It transforms an initial Gaussian noise into a more expressive distribution. The UNet is trained jointly with the rest of the model through our overall objective function. It learns to shape the prior distribution of $z$ to better match the posterior distribution inferred from trajectories and returns, but it does not directly take these as inputs. As a result, the suggested ablations of removing RTG or trajectory conditioning from the UNet are not applicable to our model.
> > Q5: Distribution over $z$: recommend to try 2D $z$
>
> Great idea! Following your suggestion, we add a new experiment using a 2D latent space ($z$ with dim=2) and apply it to the maze2d-umaze environment. This low-dimensional setting allows for easy visualization and analysis of the latent space distribution.
>
> As designed, $z_0$ is sampled from a 2D Gaussian distribution. After UNet, the distribution of $z$ is notably different from Gaussian, as seen in **Fig. 3 of our global response PDF**. This experiment shows that our model learns a non-Gaussian latent representation that captures task-relevant information. We would like to highlight that our method doesn't impose any specific distributional assumptions on $z$. The UNet is free to transform the initial Gaussian distribution into the form that is most informative for the task at hand, allowing the model to encode richer, more complex patterns about the trajectories and strategies.
>
> Thanks again for your insightful feedback and for taking the time to review our work.

---

> > ### Comment · Reviewer_W2Dk · 2024-08-10
> >
> > I thank authors for their detailed response. Most of my concerns were resolved and I'm adjusting the score.

---

> > > ### Comment · Area_Chair_mHTu · 2024-08-12
> > > **Reviewer Response Requested**
> > >
> > > Dear Reviewer,
> > >
> > > The discussion time is coming to an end soon. Please engage in the discussion process which is important to ensure a smooth and fruitful review process. Give notes on what concerns have not been addressed and why.

---

> > > ### Author Response · Authors · 2024-08-12
> > > **Thank you!**
> > >
> > > Thank you for your positive feedback and thorough review. We’re glad all your concerns have been addressed. Have a nice day!

---

### Official Review · Reviewer_NC2f · 2024-07-07

**Soundness:** 4
**Presentation:** 3
**Contribution:** 3
**Rating:** 6
**Confidence:** 3

**Summary:**

The paper introduces the Latent Plan Transformer (LPT), a novel framework for trajectory generative modeling in the absence of step-wise reward. This framework employs a top-down latent variable model, using a temporally-extended latent variable z to represent a plan for decision-making. The framework comprises three components connected by the latent plan, a neural transformation of a Gaussian noise, a transformer-based trajectory generator, and a return estimator. LPT is optimized through maximum likelihood estimation on offline datasets composed of trajectory-return pairs. During testing, a latent plan is inferred based on an expected return, after which the trajectory generator is applied to extract actions. The framework is extensively evaluated across several environments, including gym locomotion (including antmaze), kitchen, maze2d, and connectfour. LPT addresses the challenge of temporal consistency while demonstrating competitive benchmark performance compared to baselines and excelling in various aspects, spanning credit assignments, trajectory stitching, and dealing with environment contingencies.

**Strengths:**

- LPT creatively enables learning from trajectory-return pairs without any step-wise reward, while resolving the temporal consistency issue.
- The paper provides a detailed analysis from a sequential decision-making perspective, identifying the significance of plan prediction in enforcing temporal consistency.
- Exploitation-inclined Inference provides a simple and flexible way to control exploration and exploitation for latent plan sampling, which generally leads to better plans given the evaluation results.
- The method is thoroughly evaluated, and LPT-EI exhibits superior performance compared to final-return baselines and strong stitching capabilities.
- The paper is well-written and well-structured.

**Weaknesses:**

- Training and inference may suffer from inefficiency because of MCMC sampling in more complex scenarios, especially when modeling high-dimensional distribution (e.g. image-based benchmarks) or requiring high accuracy.
- As mentioned by the authors, the LPT may fail on datasets with a skewed distribution of trajectory lengths (e.g. antmaze-large) due to its sequence modeling nature.

**Questions:**

- Line 77-78, duplicate "Consider" sentences.
- What's the difference between $\bar{\theta}$ and $\theta$ in eq. 9?
- How the expected return is chosen for each dataset?
- In Figure 3, are the testing $z_0$s sampled based on unseen returns? For LPT, we need to choose an expected return before the latent plan is sampled, in many cases we may want to sample a trajectory with a return even beyond the highest one in the training data. Therefore, it would be valuable for the authors to provide an analysis of how LPT performs across a range of returns to assess its robustness when interpolating and extrapolating out-of-distribution returns.

**Limitations:**

The authors adequately discussed the limitations of the proposed method.

---

> ### Author Rebuttal · Authors · 2024-08-07
>
> Thank you for your thorough and constructive review of our paper! We shall address your questions point by point.
> > W1: Potential training and inference inefficiency because of MCMC sampling in more complex scenarios
>
> Thank you for your insightful question! We would like to clarify that with careful design, MCMC does not become a bottleneck in our method.
>
> **Sampling in low-dim latent space**: Our MCMC sampling occurs in an extremely low-dimensional latent space compared to the entire trajectory. This latent space dimensionality can be controlled independently of input data dimensionality, mitigating the curse of dimensionality often associated with MCMC in high dimensions, even for high-dimensional inputs like images.
>
> **Training Efficiency**: During training, when sampling $p(z|\tau,y)\propto p(\tau|z)p(y|z)p(z)$,  the main overhead is backpropagating through $p(\tau|z)$ over long sequences multiple times. We mitigate this by using a fixed context window in the causal Transformer (e.g., K=32) to simplify attention computations. Additionally, we adopt a persistent Markov chain (line 153) that amortizes sampling across training iterations, reducing sampling iterations to 2. This approach has proven effective, allowing us to train on complex tasks without prohibitive computational costs.
>
> **Inference Efficiency**: During inference, we don't need to backpropagate through the trajectory generator $p(\tau|z)$. As the latent plan decouples trajectory generation and return prediction (Sec 3.3), sampling only involves the much faster MLP for return prediction and the prior i.e. $p(z|y)\propto p(y|z)p(z)$, which is computationally lightweight. This makes the inference process fast.
>
> > W2: As mentioned, LPT may fail on datasets with a skewed distribution of trajectory lengths due to its sequence modeling nature.
>
> Thank you for raising this important point. The performance of sequence models like LPT and DT can indeed be affected by datasets with skewed distributions of trajectory lengths, as in the antmaze-large environment. This sensitivity to trajectory length distribution stems from the essence of sequence models, which learn to make decisions by discovering correlations between trajectories and their associated returns. When faced with a dataset that has an uneven distribution of trajectory lengths, these models may struggle to generalize effectively across time scales. Moving forward, we are keen to explore ways to address this challenge. We welcome any further thoughts or suggestions you might have on this matter.
>
> > Q1: Line 77-78, duplicate "Consider" sentences.
>
> Thank you for catching that. We will correct the duplicate sentence in the revision.
>
> > Q2: Difference between $\bar\theta$ and $\theta$ in eq. 9
>
> Thanks for raising this question. We kindly refer you to our **Global Response [Explanation on Equation 9]**: on explanations of Eq. (9). To be brief, this equation stems from the principle that maximizing likelihood is equivalent to minimizing the KL divergence between the model and data distribution. Then we can make it explicit that we have $p^y_\mathcal{D}(\tau, z) = p^y_{\bar{\theta}}(z|\tau) p^y_\mathcal{D}(\tau)$. Then we can derive,
> \begin{align*}
> D_\mathrm{KL}(p^y_\mathcal{D}(\tau, z) || p^y_\theta(\tau, z)) &=
> D_\mathrm{KL}(p^y_{\bar{\theta}}(z|\tau) p^y_\mathcal{D}(\tau) || p^y_{\theta}(z|\tau) p^y_\theta(\tau)) \\\\&=
> D_\mathrm{KL}(p^y_\mathcal{D}(\tau) || p^y_\theta(\tau))  + \mathbb{E}\_{p^y_\mathcal{D}(\tau)}D_\mathrm{KL}(p^y_{\bar{\theta}}(z|\tau) || p^y_\theta(z|\tau))
> \end{align*}
> The gradient of this KL divergence with respect to model parameters $\theta$ is consistent with Eq. 4, considering the stop_grad() operator.
>
> > Q3: Expected return selection
>
> We chose the testing expected return as the expert-level performance metrics provided by the D4RL benchmark (Fu et al., 2020). The expected returns correspond to 100 on the normalized scale in D4RL for each environment. This setting aligns with experiments in Decision Transformer.
>
> > Q4-1: Plot of the testing $z_0$s in Figure 3
>
> Yes you are correct. For the training and testing $z_0$, we refer to $p(z_0|\tau,y)$ and $p(z_0|y)$ respectively. The testing $z_0$ values in Figure 3 are sampled based on unseen returns. For the Maze2D-medium dataset, the maximum trajectory normalized score in the training data is 12.8, significantly lower than our testing return (normalized score of 100).
>
> > Q4-2: LPT performance across return ranges
>
> We thank the reviewer for this valuable suggestion. Our choice of normalized score 100 for $y$ is already out-of-distribution (OOD) for all datasets used. To illustrate this, we provide a table of D4RL dataset statistics with normalized scores:
> | Dataset| max. score|95 pct. score | 90 pct. score |
> |-|-|-|-|
> | maze2d-umaze-v1|21.1|13.2|10.3|
> | maze2d-medium-v1|12.8| 6.8|4.9|
> | maze2d-large-v1|16.9| 6.5| -2.5|
> | hopper-medium-v2 | 99.5 | 63.2|57.0|
> | hopper-medium-replay-v2| 98.6| 46.4|31.5 |
> | halfcheetah-medium-v2|45.0| 43.0|42.5|
> | halfcheetah-medium-replay-v2 |42.4|39.9|39.2|
> | walker2d-medium-v2| 92.0|83.4|82.4|
> | walker2d-medium-replay-v2|89.9|66.6|42.5|
>
> As evident from this table, our chosen $y$ is an extrapolation beyond the dataset trajectories for all environments, particularly for maze2d datasets where the max score is notably suboptimal.
>
> To further address the reviewer's concern about LPT's performance across different return ranges, we conducted an additional experiment on the Walker2D-medium-replay environment. We tested LPT with $y$ being the 50th and 90th percentiles of dataset returns, as well as the maximum dataset return and expert-level performance. As shown in **Fig. 1 of global response PDF**, LPT shows a linear trend in performance as we increase the target return. This analysis highlights LPT's robustness in both interpolating and extrapolating to OOD returns, a crucial capability for effective planning in various scenarios.
>
> Thank you again for your valuable feedback!

---

> > ### Comment · Reviewer_NC2f · 2024-08-10
> > **Rebuttal Acknowledgement**
> >
> > I would like to thank the authors for their detailed responses and additional experiments. It's very helpful to know that the expected return used for evaluation is normalized score 100. Most of my concerns and questions have been addressed, I would like to maintain my positive rating.

---

> > > ### Comment · Area_Chair_mHTu · 2024-08-12
> > > **Review Discussion Requested**
> > >
> > > Dear Reviewer,
> > >
> > > The discussion time is coming to an end soon. Please engage in the discussion process which is important to ensure a smooth and fruitful review process. Give notes on what concerns have not been addressed.

---

> > > ### Author Response · Authors · 2024-08-12
> > > **Thank you for your time!**
> > >
> > > Thank you for your positive feedback and thorough review. We’re pleased that your concerns have been addressed. Have a great day!

---

### Official Review · Reviewer_uuQw · 2024-07-13

**Soundness:** 3
**Presentation:** 3
**Contribution:** 3
**Rating:** 7
**Confidence:** 3

**Summary:**

Building on the idea of decision transformer, the paper introduces a new generative model based decision-making agent called Latent Plan Transformer (LPT). Instead of directly generating the trajectories and returns as in the prior work, LPT would first generate a latent vector, and then generate the trajectory and its return conditioned on the latent vector. The idea of introducing this latent vector is to view it as a plan which provides the agent a temporal consistency guideline in the long decision-making process. Experimental results show improved performance of LPT over existing baselines in robotic and navigation tasks.

**Strengths:**

- The idea of having a plan for decision-making is very interesting. Discussion in Section 4 on some intuitive reasons why having a plan may help in long range problems when temporal consistency is an issue.

- Experimental results show strong performance in various tasks. Especially in the maze domains with long-range delayed rewards, the performance improve from DT to LPT highlights the benefits of the latent plan.

**Weaknesses:**

- Although there are motivations and some high-level discussions provided, it would probably be more convincing if those ideas have connection to some theoretical analysis in MDP representation learning.

- The ablation study in A.5 comparing different latent prior seems important, but little discussion is provided. Intuitively, DT may be view as LPT with a trivial prior. Are there more comparisons with different priors for the latent vector? Since there is a UNet used in LPT, LPT may end up having much more parameters than DT. Are the sizes of the models adjusted to account for this extra prior network in LPT?

**Questions:**

- Are there derivations for Eq. (9)? Why is the second term in (9) are between the two parameters of the same model and one with stop_grad( )?

- Is the comparison with baselines fair in the sense that all the models have similar parameter sizes?

**Limitations:**

Yes, the authors addressed the limitations.

---

> ### Author Rebuttal · Authors · 2024-08-07
>
> Thank you for recognizing the importance of our approach to modeling plans in decision-making processes! We greatly appreciate your insightful feedback and the opportunity to clarify and expand upon our work.
>
> > W1: What are the connections between LPT and some theoretical analysis in MDP representation learning?
>
> Thank you for bringing up this interesting perspective! While there are indeed some conceptual similarities, we believe it is important to highlight a few key differences. Traditional MDP representation learning typically focuses on learning state representations that capture the MDP structure, including state transitions and reward functions, often operating on a step-by-step basis. In contrast, LPT doesn't explicitly learn the MDP structure. Instead, our latent variable $z$ serves as a policy representation, encoding information about the entire trajectory. This fundamental difference in focus - state representation versus policy representation - leads to distinct approaches. While MDP representation learning aims to facilitate efficient learning of optimal policies through compact state representations, latent variable $z$ in LPT directly informs action selection throughout the episode. This episode-level approach in LPT can be seen as a temporal abstraction over the trajectory, instead of over a timestep, as in MDP representations. Despite these differences, we acknowledge that both approaches share the goal of finding useful abstractions that facilitate decision-making in sequential problems.
>
>
> > W2.1: A.5 ablation study important but lacks discussion -- DT as LPT with a trivial prior. Are there more comparisons with different priors for the latent vector?
>
> We would like to clarify that in our LPT model, the prior is not a trivial distribution but rather a learned distribution modeled by an implicit generative model, specifically a UNet. This choice allows for a more flexible and expressive prior compared to standard distributions. Our ablation studies in A.5 demonstrate the importance of this learnable prior. For example, in the Connect Four environment, LPT with the UNet prior achieves a score of $0.99 ± 0.01$, while LPT with a standard Normal prior (without UNet) achieves $0.90 ± 0.06$. Similar improvements are observed across Gym-Mujoco tasks.
>
> Per the reviewer's suggestion, in addition to comparing learnable and non-informative priors, we performed an additional set of experiments testing the effects of different UNet priors on the model performance. The results are as follows:
>
>
> | Model prior                          | Normalized score |
> | --------                             | -------- |
> |  UNet                                | $72.31 \pm 1.92$ |
> |  UNet w/ smaller dimension           | $64.06 \pm 1.94$ |
> |  UNet w/ fewer multipliers           | $64.59 \pm 1.54$ |
> |  UNet w/ smaller initial conv        | $70.49 \pm 2.84$ |
> |  UNet w/ single ResBlock             | $67.95 \pm 4.64$ |
> |  No UNet                             | $56.88 \pm 4.20$ |
>
> Our results underscores the crucial role of the learned prior in LPT's performance. Our ablation studies further reveal that our current UNet configuration is optimal among the variants tested. We appreciate the reviewer's suggestion, as it inspires us to perform a more detailed model analysis.
>
>
>
> > W2.2 and Q2: LPT may have much more parameters than DT due to the UNet used. Are the sizes of the models adjusted to account for this extra prior network in LPT?
>
> Thank you for bringing up this important aspect in model training! Throughout our project, we have taken cautious to ensure a fair comparison between DT and LPT. We used the same implementation of the autoregressive decoder as DT, and since we don't have return-to-go conditioning in LPT, the total number of parameters end up being comparable:
>
> * DT with 3 layers and hidden size 128 has approximately 1.5 million parameters.
> * LPT with the UNet prior has around 1.6 million parameters.
>
> While there is a slight increase in the number of parameters for LPT, we believe this difference is not substantial enough to solely account for the performance improvements we observe. Importantly, both models use the same decoder implementation, ensuring a consistent base architecture for comparison. The additional parameters in LPT are primarily in the UNet-based prior, which allows for more expressive modeling of the trajectory distribution.
>
>
> > Q1: Are there derivations for Eq. (9)? Why is the second term in (9) are between the two parameters of the same model and one with stop_grad()?
>
>
> We kindly refer you to our **Global Response [Explanation on Equation 9]**: on explanations of Eq. (9). To be brief, the understanding of this equation stems from the principle that maximizing likelihood is equivalent to minimizing the KL divergence between the model and data distribution. Then we can make it explicit that in this equation, we have $p^y_\mathcal{D}(\tau, z) = p^y_{\bar{\theta}}(z|\tau) p^y_\mathcal{D}(\tau)$. Then we can derive,
>
> \begin{align}
> D_\mathrm{KL}(p^y_\mathcal{D}(\tau, z) || p^y_\theta(\tau, z)) &=
> D_\mathrm{KL}(p^y_{\bar{\theta}}(z|\tau) p^y_\mathcal{D}(\tau) || p^y_{\theta}(z|\tau) p^y_\theta(\tau)) \\\\&=
> D_\mathrm{KL}(p^y_\mathcal{D}(\tau) || p^y_\theta(\tau))  + \mathbb{E}\_{p^y_\mathcal{D}(\tau)}D_\mathrm{KL}(p^y_{\bar{\theta}}(z|\tau) || p^y_\theta(z|\tau))
> \end{align}
>
>
> The gradient of this KL divergence with respect to model parameters $\theta$ is consistent with Eq. 4, considering the stop_grad() operator.
>
> Thank you again for your insightful review!

---

> > ### Comment · Reviewer_uuQw · 2024-08-13
> >
> > Thank you for the reply. Indeed most existing MDP representation learning focuses on learning some step-by-step models, while the proposed method learns a latent variable for the entire trajectory. But from existing theories, learning such a whole trajectory variable might not be useful from the decision-making point of view, as the state itself should have included all necessary information for making optimal decisions. The benefits of the latent variable could be due to its advantages in helping state compression/aggregation. This may be beyond the scope of this paper, but having some theoretical analysis on a simple problem showing such benefits might further strengthen the claims of the paper.
> >
> > As of the ablation study and fair comparison to the baseline, thank you for the additional stats and results. I think they are important and could be highlighted a bit more in the main text.

---

> > > ### Author Response · Authors · 2024-08-14
> > > **Thank you for your insightful feedback!**
> > >
> > > Dear Reviewer,
> > >
> > > Thank you for your deeply insightful feedback. We appreciate your insight that the benefits of our latent variable modeling could stem from its advantages in state compression/aggregation. We will follow your advice to conduct theoretical analysis to strengthen our claims. We'll also highlight our ablation study and comparison results in the main text, as suggested.
> > >
> > > We're grateful for your engagement and continued support.
> > >
> > > Best,
> > >
> > > Authors.

---

> ### Comment · Area_Chair_mHTu · 2024-08-12
> **Reviewer Discussion Needed**
>
> Dear Reviewer,
>
> The discussion time is coming to an end soon. Please engage in the discussion process which is important to ensure a smooth and fruitful review process. Give notes on what parts of the reviewers responses that have and have not addressed your concerns.

---

### Author Rebuttal · Authors · 2024-08-07

## Global Response

We would like to thank all reviewers for their helpful feedback! Here, we would like to add some clarifications based on points raised by multiple reviewers.

**[Latent variable $z$ formulation]**

We first higlight that the latent variable of our model is $z$, whose prior distribution is a learnable transformation from Gaussain white noise $z_0\sim N(0,I)$. It is parameterized by an expressive UNet. Meanwhile, this prior model can be viewed as an implicit generator with implicit densities.

**[Additional explanation on Equation 9]**

To understand Eq. 9, we can start from the well-known equivalence between Maximum Likelihood Estimate $\max\_\theta\mathbb{E}\_{p^y_\mathcal{D}(\tau, z)}[\log p^y_\theta(\tau, z)]$ (Eq. 4) and KL minimization $\min\_\theta\mathbb{E}\_{p^y_\mathcal{D}(\tau, z)}[\log p^y_\mathcal{D}(\tau, z) - \log p^y_\theta(\tau, z)]$ (Eq. 9). Here $p^y_\mathcal{D}(\tau, z) = p^y_{\bar{\theta}}(z|\tau) p^y_\mathcal{D}(\tau)$. We have a stop_grad version of $\bar\theta$ because deriving Eq. 4 from the marginal MLE $\max_\theta\mathbb{E}\_{p^y_\mathcal{D}(\tau)}[\log p^y_\theta(\tau)]$ does not involve a derivative through $\mathbb{E}\_{p^y_{{\theta}}(z|\tau)}$ (as shown in Appx. A.1).

**[Comparison between decision-making algorithms]**

We hope it is helpful to further clarify on the implicit data specification in different decision-making algorithms. Sequence-based methods (e.g. DT and LPT) and classical TD-learning methods (e.g. CQL) have different assumptions in data distribution. The non-Markovian autoregressive nature of DT and LPT requires the training data to have reasonable coverage of sequence length. In contrast, CQL operate primarily on individual state-action-reward-next_state tuples, which can be advantageous in datasets with skewed trajectory length distributions.

---

### Decision · Program_Chairs · 2024-09-25

**Decision:**

Accept (poster)

**Comment:**

This paper presents a new method to improve the representations learned in decision transformers. After the rebuttal and discussion time, the reviewers agree that the paper is promising and adds to our understanding of how to improve decision transformers. There are still some outstanding questions about how this representation learning method relates to other representation learning methods in RL, and we encourage the authors to continue investigating this area.